# Factors associated with registration for organ donation among clinical nurses

Li-Chueh Weng[1,2]☯*, Yang-Jen Chiang[3,4]☯, Hsiu-Li Huang[5], Yu-Hsia Tsai[1,6], Kang-Hua Chen[1,7], Woan-Shyuan Wang[1], Mei-Hsiu Lin[3,7]

1 School of Nursing, College of Medicine, Chang Gung University, Taoyuan, Taiwan, 2 Department of General Surgery, Chang Gung Medical Foundation, Linkuo Medical Center, Taoyuan, Taiwan, 3 Division of Transplant Urology, Department of Surgery, Chang Gung Medical Foundation, Linkou Medical Center, Taoyuan, Taiwan, 4 College of Medicine, Chang Gung University, Taoyuan, Taiwan, 5 Department of Long-Term Care, College of Health Technology, National Taipei University of Nursing and Health Science, Taipei, Taiwan, 6 Department of Cardiovascular Medicine, Chang Gung Medical Foundation, Linkou Medical Center, Taoyuan, Taiwan, 7 Department of Nursing, Chang Gung Medical Foundation, Linkou Medical Center, Taoyuan, Taiwan

☯ These authors contributed equally to this work.
* ax2488@mail.cgu.edu.tw

**Data Availability Statement:** All relevant data are within the manuscript files.

**Funding:** The study was supported from the Ministry of Health and Welfare Taiwan (number

## Abstract

### Purpose

Healthcare professionals play an important role in the organ donation process. The aim of this study was to examine the organ donation registration rate and related factors among clinical nurses.

### Material and methods

In this cross-sectional, correlational study, we used mailed questionnaires to collect data from four geographical areas and three hospital levels in Taiwan from June 6 to August 31, 2018. Two thousand and thirty-three clinical nurses participated in this study.

### Results

Participants' mean age was 34.47 years, and 95.7% were women. Of them, 78.3% were willing to donate their organs and 20.6% had registered for organ donation after death. The results of logistic regression showed that in the personal domain, higher age (odds ratio (OR) = 1.03, p < 0.001), better knowledge of organ donation (OR = 1.09, p < 0.001), and a positive attitude toward organ donation (OR = 2.91, p < 0.001) were positively associated with organ donation registration, while cultural myths (OR = 0.69, p < 0.001) were negatively correlated. In the policy domain, the convenience of the registration procedure (OR = 1.45, p < 0.001) was positively associated with registration. A gap between willingness to donate and actual registration was observed.

### Conclusions

Personal factors played an important role in organ donation registration. Therefore, efforts to improve knowledge and inculcate positive cultural beliefs about organ donation among

GMRPD1H0011), The funders had no role in study design, data collection and analysis, decision to publish, or preparation of the manuscript.

**Competing interests:** The authors have declared that no competing interests exist.

clinical nurses are recommended. There is also a need to cooperate with government policies to provide appropriate in-service training and policy incentives and establish an efficient registration process.

## Introduction

Organ transplantation is one of the most important treatments for patients with end-stage diseases. However, vast demand and inadequate organ availability remain problematic. Therefore, achieving self-sufficiency by facilitating the organ donation process is an urgent goal for every country [1, 2]. Efforts to promote deceased organ donation in Taiwan include the Brain Death Legislation, Human Organ Transplantation Act, Taiwan Organ Registry and Sharing Center (TORSC), and a computer-based organ matching system. Recently, a legal framework governing the transplantation and allocation of human organs, which ensures the prioritization of waitlisted candidates if a close family member has previously donated an organ, has been implemented; this is known as the third-grade priority system [3, 4]. Owing to these efforts, Taiwan's annual deceased organ donation rate increased from 6.7 per million people in 2005 to 12.3 in 2016 [4]. However, the donation rate still has potential for improvement [5].

While deceased organ donation policies differ across countries, largely, the practice is based on two principles: opt-out and opt-in. According to the opt-out or presumed consent principle, each individual is a potential organ donor unless they have previously explicitly stated otherwise. On the contrary, as per the opt-in or normative consent principle, which is the norm in many Asian countries including Taiwan, people who wish to donate their organs after death can register on their driver's license or any official paper [6]. In Taiwan, under the opt-in principle, if there is a possibility of organ donation because of a fatal accident/emergency, the organ procurement organization or hospital must first seek the potential donor's relatives for permission. At a time of shock, this puts families in a difficult position, but being aware of the patient's wishes because of their registration status can facilitate decision-making [7]. Knowing that the potential organ donor had registered in advance could also help donation and transplantation coordinators improve communication with families and further facilitate successful organ donation. Therefore, promoting registration for organ donation is considered an important task for countries that adopt the opt-in principle.

Healthcare professionals play an important role in organ donation [8–10] and are also the fundamental link between society and the health system [11]. With regard to discussions concerning organ donation, donors and their families seek well-trained healthcare professionals who present a positive attitude toward this treatment avenue [7, 12, 13]. Studies have reported that 55–82.8% of clinical nurses and healthcare professionals support deceased organ donation and are willing to donate their organs after death [8, 9, 11, 14, 15]. Evaluating nurses' attitudes as well as willingness toward organ donation is important because they are first-line healthcare workers who continuously provide care to donors and their families [16]. Thus, more studies to reveal clinical nurses' perspectives of organ donation are need.

There have been comparatively more studies on the influencing factors of willingness than registration rates, but the results are not completely conclusive. The influencing factors of willingness toward organ donation include perceptions about organ donation [17], higher age, mistrust in the healthcare system, respect for the corpse, religious beliefs [8, 9, 11, 15], and family factors [7]. Regarding registration for deceased organ donation, higher age, lower education, lack of insurance, unemployment, comorbid conditions, and religious beliefs were

associated with lower rates [18]. Understanding the influencing factors of the registration rate would be helpful for policymaking, policy investment, and identifying groups for intervention. Therefore, the aims of this study, focused on clinical nurses, were to investigate the registration rate for organ donation and examine associated factors in this population.

## Materials and methods

### Sample and setting

This study employed a cross-sectional, correlational design. The inclusion criterion was clinical registered nurses employed in hospitals, regardless of their years of experience, gender, or specialty. Participants were recruited through simple stratified sampling based on geographic area and hospital level. As per the governmental definition, hospitals were categorized as the following: tertiary medical center, regional hospital, and local hospital level. The final 265 hospitals recruited for the sample had the following geographic distribution—north: 106, middle: 77, south: 66, and east: 16. By hospital level, they were divided as follows: 19 tertiary medical centers, 83 regional hospitals, and 163 local hospitals.

### Study variables and measurement tool

**Registration for deceased organ donation.** Participants were asked to answer two yes/no questions: one concerning their willingness toward organ donation and the second concerning whether they had registered for deceased organ donation on the National Health Insurance card.

**Factors associated with registration.** The authors reviewed the published literature and the details available on the TORSC website; this information, together with their personal experiences, was used to identify the factors associated with deceased organ donation registration. These were categorized into three domains: *personal*, *healthcare setting*, and *policy*. The content of the questionnaires is presented in S1 and S2 Files.

Personal domain factors included *demographic data*, *knowledge of organ donation*, *attitude toward organ donation*, and *cultural myths*. Demographic data included age, years of clinical experience, gender, educational level, marital status, and religion. Knowledge of organ donation included awareness of organ donation legislation as well as donation and transplantation procedure (15 items), with sample items such as "Two professional doctors are required to examine patients with end-stage disease receiving hospice care for cardiac death organ donation" and "The brain death procedure needs to be checked twice, with an interval of 12 hours." A score of 1 was assigned for a correct answer and 0 for an incorrect answer or not knowing the answer. A higher score indicated more accurate knowledge of organ donation. Attitude toward organ donation (13 items) assessed participants' perspectives, with sample items such as "Donating one's organs after death is moral and can help other people." Participants were requested to indicate their level of agreement for each item on a five-point Likert scale ranging from 1 (strongly disagree) to 5 (strongly agree). Cultural myths (five items) investigated specific beliefs about organ donation. Sample items included "It is important for the body to remain intact after death" and "Discussing deceased organ donation brings bad luck." Participants were requested to indicate their level of agreement for each item on a five-point Likert scale ranging from 1 (strongly disagree) to 5 (strongly agree). A higher score indicated a higher level of conformance with cultural myths.

The healthcare setting domain included *geographic area* (north, middle, south, and east), *hospital level* (tertiary medical center, regional hospital, and local hospital), *specialty of working unit* (general ward, emergency unit/intensive care unit, and others (e.g., outpatient department)), and the *practical difficulties encountered in organ donation* in the hospital. Practical

difficulties (nine items) concerned the hurdles in collaborating for and promoting organ dona-tion/procurement among departments. Sample items included "It is difficult to collaborate with the other departments in relation to organ procurement and donation" and "Staffing is a big issue that can make organ donation difficult." The participants were requested to indicate their level of agreement for each item on a five-point Likert scale ranging from 1 (strongly dis-agree) to 5 (strongly agree). A higher score indicated greater practical difficulties in organ donation.

The policy domain included *convenience of registration*, *feasibility of the opt-out principle*, and *third-grade priority policy*. Convenience of registration (three items) covered whether the participants agreed with the organ donation registration procedure and its accessibility. Responses were obtained on a five-point Likert scale from 1 (strongly disagree) to 5 (strongly agree). A higher score indicated greater convenience. Only one item sought information about participants' perceptions of the feasibility of the opt-out principle since it is not currently implemented in Taiwan. Another item asked if participants believed that third-grade priority has the potential to improve the organ donation rate. Responses were provided on a five-point Likert scale from 1 (strongly disagree) to 5 (strongly agree). A higher score indicated that bet-ter conformance with the policy can help improve organ donation rates.

Seven experts including three healthcare professors, one transplantation coordinator, and two nurse specialists in organ transplantation care were invited to examine the contents of the questionnaire. The content validity index was 0.90. The internal consistency reliability of each aspect (Cronbach's α) was as follows: knowledge of organ donation: 0.67; attitude toward organ donation: 0.87; cultural myths: 0.81; practical difficulties in organ donation: 0.89; and conve-nience of registration: 0.60. Thus, the questionnaire had acceptable reliability and validity.

## Ethical considerations

The Chang Gung Medical Foundation Institutional Review Board approved this study on April 24, 2018 (approval number: 201800563B0). The committee waived the need for written consent but suggested providing an explanatory cover letter with the questionnaire. The cover letter included the study purpose, procedure, and details concerning participants' personal information, such as assuring them of anonymity, confidentiality, and that the published results would contain only de-identified data. Participation was voluntary.

## Data collection

Data collection was via the mailed questionnaires. After the institutional review board approved the research proposal, TORSC staff helped us contact hospital administrators and seek their assistance with distributing the questionnaire to nurses. Then, the official research description, an explanatory cover letter, and the questionnaire were mailed to the hospital administration department. Based on the nursing capacity of each institution, 100–150 ques-tionnaires were mailed to medical centers, 20–50 to regional hospitals, and 5–10 to local hospi-tals. In order to increase the response rate, each questionnaire included a gift certificate worth US $3.5 as a token of gratitude. After completion, nurses were asked to return the question-naires by mail in pre-stamped envelopes.

## Data analysis

Data were analyzed using SPSS version 22 (IBM Corp., Armonk, NY, USA). Descriptive statis-tics were used to estimate central tendency (mean) and spread (standard deviation) for contin-uous data such as age and years of clinical experience, while frequencies and percentages were used for categorical data including gender, educational level, and marital status. The chi-

square test was used to examine the associations between categorical variables and registration status (registered or unregistered). The independent samples t-test was used to examine the differences in continuous variables between registered and unregistered participants. To analyze important factors in the context of organ donation registration, first, univariate logistic regression was conducted to analyze the effect of variables that showed significant differences in the t-test and chi-square test. Variables that were statistically significant in the univariate logistic regression ($p < 0.05$) were then included together in the multivariate logistic regression to examine their associations with organ donation registration. The significance level was set at $p < .05$.

## Results

During the study period from June 6, 2018, to August 31, 2018, there were 2358 questionnaires distributed and 2064 returned. Of these, 31 questionnaires were excluded owing to incomplete responses. Finally, 2033 questionnaires were analyzed (actual response rate 86.2%). The demographic characteristics of the sample were comparable to those of employed nurses in Taiwan. Of the 2033 participants, 1592 (78.3%) were willing to donate their organs after death, and 419 (20.6%) had registered for organ donation on their National Health Insurance cards. There were significant differences in age, years of clinical experience, and religion between the groups. The registered group had higher knowledge of organ donation, believed in fewer cultural myths, and had a more positive attitude toward organ donation than the unregistered group (Table 1).

In the healthcare setting domain, no variables showed statistically significant differences (Table 2).

In the policy domain, compared to those who were unregistered, the registered group considered the registration procedure more convenient ($t = -11.20$, $p < 0.001$), the opt-out

**Table 1. Comparison of personal data between the registered and unregistered groups (N = 2033).**

| Variables | Category | Total | Registered | | Unregistered | | χ2/t | p |
|---|---|---|---|---|---|---|---|---|
| | | | (n = 419) | | (n = 1614) | | | |
| | | Mean (SD) n (%) | Mean (n) | SD (%) | Mean (n) | SD (%) | | |
| Age | | 34.47 (8.39) | 36.49 | 8.56 | 33.95 | 8.27 | -5.53 | < 0.001 |
| Clinical experience (years) | | 10.68 (7.66) | 12.26 | 8.07 | 10.26 | 7.49 | -4.27 | 0.007 |
| Gender (n = 2025) | Women | 1937 (95.7) | 404 | 20.9 | 1533 | 79.1 | 2.27 | 0.13 |
| | Men | 88 (4.3) | 12 | 13.6 | 76 | 86.4 | | |
| Education (n = 2031) | Under college | 505 (24.9) | 97 | 19.2 | 408 | 80.8 | 0.67 | 0.41 |
| | Above college | 1526 (75.1) | 321 | 21.0 | 1205 | 79.0 | | |
| Marital status | Single | 1011 (49.8) | 195 | 19.3 | 816 | 8.7 | 2.06 | 0.15 |
| (n = 2031) | Married | 1020 (50.2) | 224 | 22.0 | 796 | 78.0 | | |
| Religion (n = 2020) | No | 562 (27.8) | 122 | 21.7 | 440 | 78.3 | 14.32 | 0.006 |
| | Buddhist | 383 (19) | 99 | 25.8 | 284 | 74.2 | | |
| | Daoist | 860 (42.6) | 150 | 17.4 | 710 | 82.6 | | |
| | Christian | 144 (7.1) | 36 | 25.0 | 108 | 75 | | |
| | Other | 71 (3.5) | 12 | 16.9 | 59 | 83.1 | | |
| Knowledge | | 9.09 (2.72) | 9.79 | 2.61 | 8.91 | 2.72 | -5.98 | < 0.001 |
| Cultural myths | | 2.59 (0.77) | 2.26 | 0.75 | 2.68 | 0.75 | 10.09 | < 0.001 |
| Attitude | | 3.34 (0.48) | 3.63 | 0.45 | 3.27 | 0.47 | -14.02 | < 0.001 |

SD, standard deviation.

**Table 2. Comparison of healthcare setting data between registered and unregistered groups (N = 2033).**

| Variables | Category | Total | Registered (n = 419) | | Unregistered (n = 1614) | | χ2/t | p |
|---|---|---|---|---|---|---|---|---|
| | | Mean (SD) n (%) | Mean (n) | SD (%) | Mean (n) | SD (%) | | |
| Area | Northern | 868 (42.7) | 199 | 22.9 | 669 | 77.1 | 7.21 | 0.07 |
| | Middle | 537 (26.4) | 101 | 18.8 | 436 | 81.2 | | |
| | Southern | 549 (27.0) | 99 | 18.0 | 450 | 82.0 | | |
| | Eastern | 79 (3.9) | 20 | 25.3 | 59 | 74.7 | | |
| Level | Tertiary medical center | 589 (29) | 118 | 20.0 | 471 | 80.0 | 1.48 | 0.48 |
| | Regional | 857 (42.2) | 170 | 19.8 | 687 | 80.2 | | |
| | Local | 587 (28.9) | 131 | 22.3 | 456 | 77.7 | | |
| Setting | General ward | 1279 (62.9) | 252 | 19.7 | 1027 | 80.3 | 2.68 | 0.26 |
| | ICU/ER | 498 (24.5) | 105 | 21.1 | 393 | 78.9 | | |
| | Other | 256 (126) | 62 | 24.2 | 194 | 75.8 | | |
| Practical difficulty | | 2.95 (0.73) | 2.93 | 0.77 | 2.96 | 0.71 | 0.69 | 0.49 |

SD, standard deviation; ICU, intensive care unit, ER, emergency room.

principle more feasible (t = -2.00, p = 0.05), and the third-grade priority policy more effective (t = -7.20 p < 0.001) (Table 3).

Univariate logistic regression was conducted for each variable that was significantly different between the registered and unregistered groups. The results are presented in Table 4.

Multivariate logistic regression (forced entry model) was used to examine the factors associated with registered or unregistered status. The independent variables were statistically significant in the univariate analysis, except years of clinical experience (its high correlation with age could cause multicollinearity). The results revealed that, in the personal domain, age (odds ratio (OR) 1.03, 95% confidence interval (CI) 1.01–1.04), knowledge of organ donation (OR 1.09, 95% CI 1.05–1.15), and attitude toward organ donation (OR 2.91, 95% CI 2.05–4.12) were positively associated with registration, while cultural myths (OR 0.69, 95% CI 0.57–0.82) were negatively associated. In the policy domain, convenience was positively associated with registration (OR 1.45, 95% CI 1.19–1.78) (Table 5).

## Discussion

The aim of this study was to reveal the factors associated with deceased organ donation registration in clinical nurses. The results could increase the available knowledge regarding this issue and serve as a reference for policymaking and in-service training about organ donation.

**Table 3. Comparison of policy-level variables between the registered and unregistered groups (N = 2033).**

| Variables | Total | | Registered (n = 419) | | Unregistered (n = 1614) | | t | p |
|---|---|---|---|---|---|---|---|---|
| | Mean | SD | Mean | SD | Mean | SD | | |
| Convenience | 3.49 | 0.72 | 3.83 | 0.71 | 3.40 | 0.69 | -11.20 | < 0.001 |
| Opt-out | 3.17 | 0.90 | 3.26 | 1.00 | 3.15 | 0.87 | -2.00 | 0.05 |
| Third-grade priority | 3.77 | 0.96 | 4.07 | 0.98 | 3.69 | 0.95 | -7.02 | < 0.001 |

SD, standard deviation.

**Table 4. Summary of univariate logistic regression results (N = 2033).**

| Variables | | B | SE | Exp (B) | 95% CI | p |
|---|---|---|---|---|---|---|
| Age | | 0.04 | 0.01 | 1.04 | 1.02–1.05 | < 0.001 |
| Clinical experience | | 0.03 | 0.01 | 1.03 | 1.02–1.05 | < 0.001 |
| Religion | No (Ref.) | | | | | |
| | Buddhist | 0.24 | 0.16 | 1.26 | 0.93–1.70 | 0.14 |
| | Daoist | -0.27 | 0.14 | 0.77 | 0.58–0.99 | 0.04 |
| | Christian | 0.18 | 0.22 | 1.20 | 0.78–1.84 | 0.39 |
| | Others | -0.31 | 0.33 | 0.73 | 0.38–1.40 | 0.35 |
| Knowledge | | 0.13 | 0.02 | 1.14 | 1.09–1.19 | < 0.001 |
| Cultural myths | | -0.75 | 0.08 | 0.47 | 0.41–0.55 | < 0.001 |
| Attitude | | -1.12 | 0.13 | 5.56 | 2.28–7.22 | < 0.001 |
| Convenience | | 0.86 | 0.08 | 2.39 | 2.02–2.79 | < 001 |
| Opt-out | | 0.13 | 0.06 | 1.14 | 1.01–1.29 | 0.03 |
| Third-grade priority | | 0.46 | 0.06 | 1.58 | 1.39–1.79 | < 0.001 |

Of the participants, 20.6% had registered for organ donation. Nurses who were older, had greater knowledge of organ donation, held more positive attitudes toward organ donation, did not believe very strongly in cultural myths, and perceived the registration procedure as convenient were most likely to register for organ donation.

The registration rate for organ donation in clinical nurses was higher than in the general Taiwanese population. As of 2020, nearly 450000 persons have successfully registered for organ donation on the National Health Insurance card [19]; accordingly, rough estimates of registration rates are 1.98% and 2.92% for the total population and the 20–65 age group, respectively. However, in the present study, the figure was lower than a previous study that reported a registration rate of 47% [18]. The difference could be a result of variations in cultural backgrounds and organ donation policies across countries [2].

The association of age with deceased organ donation registration varies. In the general population, older people, that is, those aged 50 years and above, might not consider registering for deceased organ donation because of poor health [18, 20]. However, in this study, older nurses

**Table 5. Factors associated with registration status in clinical nurses (N = 2033).**

| Variables | | B | SE | Exp (B) | 95% CI | p |
|---|---|---|---|---|---|---|
| Age | | 0.03 | 0.01 | 1.03 | 1.01–1.04 | < 0.001 |
| Religion | No (Ref.) | | | | | |
| | Buddhist | 0.24 | 0.18 | 1.27 | 0.90–1.80 | 0.17 |
| | Daoist | -0.24 | 0.15 | 0.79 | 0.59–1.06 | 0.12 |
| | Christian | 0.01 | 0.24 | 1.01 | 0.63–1.63 | 0.96 |
| | Others | -0.42 | 0.38 | 0.66 | 0.31–1.37 | 0.26 |
| Knowledge | | 0.09 | 0.03 | 1.09 | 1.05–1.15 | < 0.001 |
| Cultural myths | | -0.38 | 0.09 | 0.69 | 0.57–0.82 | < 0.001 |
| Attitude | | 1.07 | 0.18 | 2.91 | 2.05–4.12 | < 0.001 |
| Convenience | | 0.37 | 0.10 | 1.45 | 1.19–1.78 | < 0.001 |
| Opt-out | | 0.03 | 0.07 | 1.03 | 0.90–1.17 | 0.68 |
| Third-grade priority | | 0.04 | 0.07 | 1.04 | 0.90–1.19 | 0.59 |
| Constant | | -7.45 | 0.76 | 0.001 | | < 0.001 |

SE, standard error, CI, confidence interval.

were inclined to register for deceased organ donation. Older clinical nurses have more experience in clinical care than younger nurses, which may influence their attitude and behavior concerning organ donation [10]. It should be noted that this may simply have been a result of the large sample size, since the OR was small. The association of age with organ donation registration may need further investigation.

Similar to previous studies, while the participants' knowledge of organ donation was not very satisfactory, it was significantly positively associated with organ donation registration [8, 9, 14]. Although the coefficient of knowledge of organ donation was small, knowledge is the basis of practice. Thus, efforts to increase clinical nurses' knowledge related to organ donation should be included in the in-service training. Our participants had a positive attitude toward organ donation, which is consistent with a previous finding that positive attitude is an important factor in organ donation registration [13]. When individuals have a positive attitude toward organ donation, they are more willing to take real actions to execute their decision [21].

Generally, cultural myths are negatively associated with organ donation registration. While in this sample there did not appear to be a dominant cultural myth, beliefs such as talking about organ donation registration bringing bad luck, respect for the corpse and the need to keep it intact, and fear about the organ recovery procedure were mentioned. Fear about the surgical recovery procedure may be rooted in misunderstandings about deceased organ donation [13–15, 17]. The belief that one's body should remain intact after death is common in Asian cultures [22, 23]. Further, shielding one's parents' bodies from harm is considered an expression of filial piety [24, 25]. Donating vital organs is considered to result in one's inability to "survive" in the world after death [13, 24]. Some Buddhists believe that the soul needs at least 24 hours to integrate and prepare to leave a deceased body, and that harm done to the body after physical death can still be "felt" in the soul; therefore, during this time, the body must not be tampered with [26, 27]. When a person is alive, it is taboo to talk about the possibility of their subsequently becoming a deceased organ donor as it makes people uncomfortable. Thus, government and healthcare institutions should work closely with opinion leaders and former donors' families to provide meaning, value, and inspiration for organ donation. Simultaneously, in aspects of donor care, intact recovery of the donor's body, traditional funeral rituals, and grief care for families should be addressed in the care guidelines for deceased organ donation.

Previous reports have claimed that religious beliefs are associated with the willingness toward organ donation [8, 9, 15]. In the current study, religion did not display a significant association in the multivariate analysis. This could be because we only sought information about the religion followed and not about the engagement with or importance of religious beliefs for the participant. In addition, religious beliefs are closely related to cultural beliefs, because of which the explanation for the religious belief may be included in the "cultural myths" variable. The influence of religious beliefs on organ donation may be complex and fail to exhibit a homogenous pattern [26]. We suggest that future studies use suitable scales or questionnaires to assess religious beliefs and explore their influence on deceased organ donation.

The majority of the participants viewed the registration procedure as convenient and thought of this as a facilitating factor. It is understandable that the registration rate will increase if the procedure is convenient and not time-consuming. Currently, people can send an application form via the hospital front desk as well as the websites of two organizations in Taiwan. It will also be convenient if registration service desks are set up in places with public access. In terms of the other two policies, those who believed that opt-out can be implemented in Taiwan and who agreed with the priority of organ allocation were more likely to register,

but the association was insignificant in the multivariate analysis. This could be explained by the other variables in the regression analysis. Notably, higher deceased organ donation rates have been observed in countries that implement the opt-out rather than the opt-in principle [28]. However, the opt-out principle may not be suitable for all countries. Before considering its implementation, cultural and ethical issues need to be addressed and citizens must achieve a consensus about organ donation after death. Third-grade priority can be said to be a pioneering initiative in Taiwan. As per this concept, donors' good deeds may not only help strangers but also benefit their loved ones. This policy may also help individuals discuss organ donation with their family members. Some scholars have noted that people who are willing to communicate with family about organ donation are more willing to donate their organs [20, 29].

The gap between the willingness to donate and actual registration remains an issue as in the general population [13, 18]. The government could use policy incentives to encourage hospitals to improve knowledge of and attitudes toward organ donation, understand the effect of cultural background on clinical nurses and other healthcare professionals, and integrate knowledge into clinical care. For example, funded hospitals could hold in-service training workshops and seminars. The core curriculum should include not only knowledge of organ donation and transplantation but also case scenarios and focus groups with religious and cultural scholars and experts to discuss the diversity of religious beliefs and cultural thinking pertaining to organ donation.

A strength of this study is its large sample size, allowing the results to be representative of clinical nurses from different regions and hospital levels. The characteristics of the sample are also consistent with those of practicing nurses in Taiwan. However, a weakness of this study is that the influence of healthcare setting factors could not be confirmed. Given its importance in promoting organ donation, the effects of factors related to the healthcare setting on registration need further investigation. In addition, while the reliability and validity of the questionnaire, which was developed for the purpose of this study, were confirmed, they require further examination.

## Conclusion

There was a gap between willingness toward deceased organ donation and the actual registration rate in clinical nurses. The results also revealed that personal domain factors are the most important with regard to organ donation registration. We recommend that future studies employ a qualitative approach to gain a comprehensive understanding of the experiences and context of clinical nurses. Further, collaborative efforts between healthcare managers and policymakers can improve the deceased organ donation rate.

## Supporting information

**S1 File. English questionnaire.**
(DOCX)

**S2 File. Traditional Chinese questionnaire.**
(PDF)

## Acknowledgments

We would like to thank all the participants who completed the questionnaires. We also sincerely acknowledge the administration department of each hospital for their help.

## Author Contributions

**Conceptualization:** Li-Chueh Weng, Yang-Jen Chiang, Hsiu-Li Huang, Yu-Hsia Tsai, Kang-Hua Chen, Woan-Shyuan Wang.

**Data curation:** Li-Chueh Weng, Hsiu-Li Huang, Yu-Hsia Tsai, Woan-Shyuan Wang, Mei-Hsiu Lin.

**Formal analysis:** Li-Chueh Weng, Yang-Jen Chiang.

**Investigation:** Li-Chueh Weng, Hsiu-Li Huang, Kang-Hua Chen, Woan-Shyuan Wang, Mei-Hsiu Lin.

**Methodology:** Yang-Jen Chiang, Yu-Hsia Tsai, Kang-Hua Chen.

**Writing – original draft:** Li-Chueh Weng, Yang-Jen Chiang, Hsiu-Li Huang, Yu-Hsia Tsai, Kang-Hua Chen, Woan-Shyuan Wang, Mei-Hsiu Lin.

**Writing – review & editing:** Li-Chueh Weng.

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
