## [Decision Letter · Decision Letter 0]

30 Oct 2020

PONE-D-20-30483

Factors associated with the registration of organ donation among clinical nurses

PLOS ONE

Dear Dr. Weng,

Thank you for submitting your manuscript to PLOS ONE. After careful consideration, we feel that it has merit but does not fully meet PLOS ONE’s publication criteria as it currently stands. Therefore, we invite you to submit a revised version of the manuscript that addresses the points raised during the review process.

The points of the reivewers are all very relevant and I expect you address them all.

We look forward to receiving your revised manuscript.

Kind regards,

Nicola Lacetera

Academic Editor

PLOS ONE

Journal Requirements:

2. For more information on PLOS ONE's expectations for statistical reporting, please see https://journals.plos.org/plosone/s/submission-guidelines.#loc-statistical-reporting. Please update your Methods and Results sections accordingly.

3. Please include additional information regarding the survey or questionnaire used in the study and ensure that you have provided sufficient details that others could replicate the analyses. For instance, if you developed a questionnaire as part of this study and it is not under a copyright more restrictive than CC-BY, please include a copy, in both the original language and English, as Supporting Information.  If the original language is written in non-Latin characters, for example Amharic, Chinese, or Korean, please use a file format that ensures these characters are visible.

4.Thank you for stating the following financial disclosure:

 [The funders had no role in study design, data collection and analysis, decision to publish, or preparation of the manuscript].

Reviewers' comments:

Reviewer's Responses to Questions

**Comments to the Author**

1. Is the manuscript technically sound, and do the data support the conclusions?

Reviewer #1: Partly

Reviewer #2: Partly

2. Has the statistical analysis been performed appropriately and rigorously? 

Reviewer #1: Yes

Reviewer #2: No

3. Have the authors made all data underlying the findings in their manuscript fully available?

Reviewer #1: Yes

Reviewer #2: No

4. Is the manuscript presented in an intelligible fashion and written in standard English?

Reviewer #1: Yes

Reviewer #2: Yes

5. Review Comments to the Author

Reviewer #1: Dear res respectable authors

Thank you for considering a good area of research. Your paper is well-designed but need some modification. I hope my comments will improve the quality of your work.

- Please prepare a structured abstract (purpose, materials and methods, results, conclusion and keywords). Also add some demographic information in the results section such as mean age and % of male or female.

- Please update your bibliography citation in accordance with guideline of journal.

- Page 4, lines 106-108, it is unclear and need revision.

- Please attach your questionnaire as a supplementary file. IN my opinion, your well-designed questionnaire can help authors from other countries to conduct similar research in this field and promote the literature.

- Please clear your sampling method.

- The mentioned response rate is not in accordance with the literature but the reason may be giving gifts to participants.

- Page 9, results section, paragraph one is lengthy. Please delete the subject that mentioned in table 1.

- Please add a paragraph about the strength , and weakness or limitation of the study in discussion section.

Cheers

Reviewer #2: Thanks authors for investigating such an important subject; there are two problems in the statistical analysis which I think need to be revised in the article.

1- Very large sample size increases the power of the study so much that even a subtle effect, rather than a practical effect, can become significant. It is the case in this study with a sample size equal to 2033, the very trivial effect of the factors such as age (OR=1.03) and knowledge of organ donation (OR=1.09) become statistically significant without having a practical interpretation. Therefore, the authors should consider the effect of the large sample size in the interpretation of the results.

2- Table 1 shows a significant relationship between religion and registration of organ donation. As the authors stated in the discussion, the non-significant effect of the religion in the logistic regression model could be as a result of multicollinearity with the cultural myths. The crude regression must be performed on each variable separately before entering the variables into the multiple regression. This way, the effect of each variable can be studied separately and it becomes clear whether the variable does not affect the outcome or its nonsignificant effect is due to collinearity with other variables in the study.

6. PLOS authors have the option to publish the peer review history of their article (what does this mean?). If published, this will include your full peer review and any attached files.

Reviewer #1: **Yes: **Morteza Arab-Zozani

Reviewer #2: No

---

## [Author Response · Author response to Decision Letter 0]

19 Nov 2020

Dear editor and reviewers: 

Thank you very much for your recommendation. We have revised the abstract, method, results, discussion, and other sections according to your comments. Please let us know whether there is anything else that is needed. 

Sincerely, 

Authors

PONE-D-20-30483

Factors associated with the registration of organ donation among clinical nurses

1. For more information on PLOS ONE's expectations for statistical reporting, please see https://journals.plos.org/plosone/s/submission-guidelines.#loc-statistical-reporting. Please update your Methods and Results sections accordingly.

Response: We have updated the statistical reporting in the methods and results section. In Data analysis section, “Descriptive statistics were used to estimate central tendency (mean) and spread (standard deviation) for continuous data such as age and years of clinical experience, while frequencies and percentages were used for categorical data including gender, educational level, and marital status. The chi-square test was used to examine the associations between categorical variables and registration status (registered or unregistered). The independent samples t-test was used to examine the differences in continuous variables between registered and unregistered participants. To analyze important factors in the context of organ donation registration, first, univariate logistic regression was conducted to analyze the effect of variables that showed significant differences in the t-test and chi-square test. Variables that were statistically significant in the univariate logistic regression (p < 0.05) were then included together in the multivariate logistic regression to examine their associations with organ donation registration.”

2.Please include additional information regarding the survey or questionnaire used in the study and ensure that you have provided sufficient details that others could replicate the analyses..

Response: We have attached the questionnaires.

3.Thank you for stating the following financial disclosure:

 [The funders had no role in study design, data collection and analysis, decision to publish, or preparation of the manuscript]. Please include your amended statements within your cover letter; we will change the online submission form on your behalf.

Response: We have included the statement within the cover letter.

Review Comments to the Author

Reviewer #1: 

1.Please prepare a structured abstract (purpose, materials and methods, results, conclusion and keywords). Also add some demographic information in the results section such as mean age and % of male or female.

Response and revision: Thank you for the recommendation. We have revised the abstract as “Purpose: Healthcare professionals play an important role in the organ donation. The aims of this study was to examine the registration rate of organ donation and its related factors among clinical nurses. 

Material and methods: This cross-sectional and correlational study using mailed questionnaires to collected data from four geographical areas and three hospital levels in Taiwan from June 6 2018 to August 31, 2018. Two thousand and thirty-three clinical nurses participated in this study.

Results: Participants’ mean age was 34.47 years, and 95.7% were women. Of them, 78.3% were willing to donate their organs and 20.6% had registered for organ donation after death. The results of logistic regression showed that in the personal domain, older in age (OR (odds ratio) = 1.03, p < 0.001), better knowledge of organ donation (OR = 1.09, p < 0.001), and a positive attitude toward organ donation (OR = 2.91, p < 0.001) were positive factors associated with organ donation registration, while cultural myths (OR = 0.69, p < 0.001) were negatively correlated. In the policy domain, convenience of registration procedure (OR = 1.45, p < 0.001) was positive factor associated with registration. A gap between the willingness to donation and real registered was happened among clinical nurses. 

Conclusions: The personal domain related factors were important factors related to the registration of organ donation. Improving knowledge and inculcate positive cultural beliefs about organ donation among clinical nurses was recommend. Hospital managers also suggested cooperate with government policies to provide appropriate in-service training, policy incentives, and establish an efficient registration process. 

Key words: attitude, culture, knowledge, nurse, organ donation, policy making”

2.Please update your bibliography citation in accordance with guideline of journal.

Response and revision: Thank you for your recommendation. We have revised the citation.

3. Page 4, lines 106-108, it is unclear and need revision.

Response and revision: Thank you for your recommendation, we have revised as “The authors reviewed the published literature and the details available on the TORSC website; this information, together with their personal experiences, was used to identify the factors associated with deceased organ donation registration. These factors were categorized into three domains: personal, healthcare setting, and policy. The content of the questionnaires is presented in S1 and S2.”

4. Please attach your questionnaire as a supplementary file. In my opinion, your well-designed questionnaire can help authors from other countries to conduct similar research in this field and promote the literature. 

Response and revision: Thank you for the recommendation. We have attached the questionnaire as supplementary file.

5. Please clear your sampling method.

Response and revision: Thank you for your recommendation, in sample section, we have added “Participants were recruited through simple stratified sampling based on geographic area and hospital level.”

6.The mentioned response rate is not in accordance with the literature but the reason may be giving gifts to participants.

Response: in data collection section we have mentioned that “In order to increase the response rate, each questionnaire included a gift certificate worth US $3.5 as a token of gratitude.” 

7.Page 9, results section, paragraph one is lengthy. Please delete the subject that mentioned in table 1.

Response and revision: Thank you for your recommendation. We have shortened the paragraph one as “The demographic characteristics of the sample were comparable to those of employed nurses in Taiwan. Of the 2033 participants, 1592 (78.3%) were willing to donate their organs after death, and 419 (20.6%) had registered for organ donation on their National Health Insurance cards. There were significant differences in age, years of clinical experience, and religion between the groups. The registered group had higher knowledge of organ donation, believed in fewer cultural myths, and had a more positive attitude toward organ donation than the unregistered group.”

8. Please add a paragraph about the strength, and weakness or limitation of the study in discussion section.

Response and revision: Thank you for your recommendation. We have added a paragraph in discussion section “A strength of this study is its large sample size, allowing the results to be representative of clinical nurses from different regions and hospital levels. The characteristics of the sample are also consistent with those of practicing nurses in Taiwan. However, a weakness of this study is that the influence of healthcare setting factors could not be confirmed. Given its important role in promoting organ donation, the effects of factors related to the healthcare setting on registration need further investigation. In addition, while the reliability and validity of the questionnaire, which was developed for the purpose of this study, were confirmed, they require further examination.”

Reviewer #2: 

1- Very large sample size increases the power of the study so much that even a subtle effect, rather than a practical effect, can become significant. It is the case in this study with a sample size equal to 2033, the very trivial effect of the factors such as age (OR=1.03) and knowledge of organ donation (OR=1.09) become statistically significant without having a practical interpretation. Therefore, the authors should consider the effect of the large sample size in the interpretation of the results.

Response and revision: Thank you for your recommendation, in discussion section, paragraph three, line257-259, we have revised as “. It should be noted that this may simply have been a result of the large sample size, since the OR was small. The association of age with organ donation registration may need more investigation” and line 262-263, “Although the coefficient of knowledge of organ donation was small, knowledge is the basis of practice.”

2- Table 1 shows a significant relationship between religion and registration of organ donation. As the authors stated in the discussion, the non-significant effect of the religion in the logistic regression model could be as a result of multicollinearity with the cultural myths. The crude regression must be performed on each variable separately before entering the variables into the multiple regression. This way, the effect of each variable can be studied separately and it becomes clear whether the variable does not affect the outcome or its nonsignificant effect is due to collinearity with other variables in the study.

Response and revision: Thank you for your recommendation, we have performed the crude regression for each variable. In the results section, line 220-222, “Univariate logistic regression was conducted for each variable that was significantly different between the registered and unregistered groups. The results are presented in Table 4.”

---

## [Decision Letter · Decision Letter 1]

17 Dec 2020

PONE-D-20-30483R1

Factors associated with the registration of organ donation among clinical nurses

PLOS ONE

Dear Dr. Weng,

Thank you for submitting your manuscript to PLOS ONE. After careful consideration, we feel that it has merit but does not fully meet PLOS ONE’s publication criteria as it currently stands. Therefore, we invite you to submit a revised version of the manuscript that addresses the points raised during the review process.

The reviewers have both "signed off" by recommending acceptance. I will condition my final acceptance recommendation to a full proof reading of the manuscript to ensure that the English flows better than it does now.

We look forward to receiving your revised manuscript.

Kind regards,

Nicola Lacetera

Academic Editor

PLOS ONE

Reviewers' comments:

Reviewer's Responses to Questions

**Comments to the Author**

1. If the authors have adequately addressed your comments raised in a previous round of review and you feel that this manuscript is now acceptable for publication, you may indicate that here to bypass the “Comments to the Author” section, enter your conflict of interest statement in the “Confidential to Editor” section, and submit your "Accept" recommendation.

Reviewer #1: All comments have been addressed

Reviewer #2: All comments have been addressed

2. Is the manuscript technically sound, and do the data support the conclusions?

Reviewer #1: Yes

Reviewer #2: Yes

3. Has the statistical analysis been performed appropriately and rigorously? 

Reviewer #1: Yes

Reviewer #2: Yes

4. Have the authors made all data underlying the findings in their manuscript fully available?

Reviewer #1: Yes

Reviewer #2: Yes

5. Is the manuscript presented in an intelligible fashion and written in standard English?

Reviewer #1: Yes

Reviewer #2: Yes

6. Review Comments to the Author

Reviewer #1: Dear Authors

Thank you for addressing my comments in a clear matter. In my opinion the quality of your article is increased and well sound.

Cheers

Reviewer #2: (No Response)

7. PLOS authors have the option to publish the peer review history of their article (what does this mean?). If published, this will include your full peer review and any attached files.

Reviewer #1: **Yes: **Morteza Arab-Zozani

Reviewer #2: **Yes: **Farzaneh Amanpour

---

## [Author Response · Author response to Decision Letter 1]

29 Dec 2020

Response and revision

Comment from the editor 

I will condition my final acceptance recommendation to a full proof reading of the manuscript to ensure that the English flows better than it does now.

Response: The author appreciate the opportunity to edit our manuscript to improve its flow. Our responses to your comment are below.

Abstract 

(page 2, line 24-43)

Purpose: Healthcare professionals play an important role in the organ donation process. The aim of this study was to examine the organ donation registration rate and related factors among clinical nurses.

Material and methods: In this cross-sectional, correlational study, we used mailed questionnaires to collect data from four geographical areas and three hospital levels in Taiwan from June 6 to August 31, 2018. Two thousand and thirty-three clinical nurses participated in this study.

Results: Participants’ mean age was 34.47 years, and 95.7% were women. Of them, 78.3% were willing to donate their organs and 20.6% had registered for organ donation after death. The results of logistic regression showed that in the personal domain, higher age (odds ratio (OR) = 1.03, p < 0.001), better knowledge of organ donation (OR = 1.09, p < 0.001), and a positive attitude toward organ donation (OR = 2.91, p < 0.001) were positively associated with organ donation registration, while cultural myths (OR = 0.69, p < 0.001) were negatively correlated. In the policy domain, the convenience of the registration procedure (OR = 1.45, p < 0.001) was positively associated with registration. A gap between willingness to donate and actual registration was observed. 

Conclusions: Personal factors played an important role in organ donation registration. Therefore, efforts to improve knowledge and inculcate positive cultural beliefs about organ donation among clinical nurses are recommended. There is also a need to cooperate with government policies to provide appropriate in-service training and policy incentives and establish an efficient registration process.

Introduction 

(page 3, line57)

However, the donation rate still has potential for improvement

(page3, Line 66-67)

At a time of shock, this puts families in a difficult position, but being aware of the patient’s wishes because of their registration status can facilitate decision-making

(page4, line 70-71)

Therefore, promoting registration for organ donation is considered an important task for countries that adopt the opt-in principle.

(page 4, line 89-90)

Therefore, the aims of this study, focused on clinical nurses, were to investigate the registration rate for organ donation and examine associated factors in this population.

Sample and setting 

(page5, line 93-100)

This study employed a cross-sectional, correlational design. The inclusion criterion was clinical registered nurses employed in hospitals, regardless of their years of experience, gender, or specialty. Participants were recruited through simple stratified sampling based on geographic area and hospital level. As per the governmental definition, hospitals were categorized as the following: tertiary medical center, regional hospital, and local hospital level. The final 265 hospitals recruited for the sample had the following geographic distribution—north: 106, middle: 77, south: 66, and east: 16. By hospital level, they were divided as follows: 19 tertiary medical centers, 83 regional hospitals, and 163 local hospitals.

Study variables and measurement tool

(page5, line 103-105)

Participants were asked to answer two yes/no questions: one concerning their willingness toward organ donation and the second concerning whether they had registered for deceased organ donation on the National Health Insurance card.

Ethical consideration 

(page 7, Line 159-162)

The cover letter included the study purpose, procedure, and details concerning participants’ personal information, such as assuring them of anonymity, confidentiality, and that the published results would contain only de-identified data. Participation was voluntary.

Data collection 

(page 8, line 164-171)

Data collection was via the mailed questionnaires. After the institutional review board approved the research proposal, TORSC staff helped us contact hospital administrators and seek their assistance with distributing the questionnaire to nurses. Then, the official research description, an explanatory cover letter, and the questionnaire were mailed to the hospital administration department. Based on the nursing capacity of each institution, 100–150 questionnaires were mailed to medical centers, 20–50 to regional hospitals, and 5–10 to local hospitals. In order to increase the response rate, each questionnaire included a gift certificate worth US $3.5 as a token of gratitude. After completion, nurses were asked to return the questionnaires by mail in pre-stamped envelopes.

Results (page 12, line 220-221)

Multivariate logistic regression (forced entry model) was used to examine the factors associated with registered or unregistered status.

Discussion 

(page 13, line 237-240)

Nurses who were older, had greater knowledge of organ donation, held more positive attitudes toward organ donation, did not believe very strongly in cultural myths, and perceived the registration procedure as convenient were most likely to register for organ donation.

(Page 13, line 245-247)

However, in the present study, the figure was lower than a previous study that reported a registration rate of 47% [18]. The difference could be a result of variations in cultural backgrounds and organ donation policies across countries [2].

(page 13-14, line 248-253)

The association of age with deceased organ donation registration varies. In the general population, older people, that is, those aged 50 years and above, might not consider registering for deceased organ donation because of poor health [18, 20]. However, in this study, older nurses were inclined to register for deceased organ donation. Older clinical nurses have more experience in clinical care than younger nurses, which may influence their attitude and behavior concerning organ donation [10].

(page 14, line 264-267)

Generally, cultural myths are negatively associated with organ donation registration. While in this sample there did not appear to be a dominant cultural myth, beliefs such as talking about organ donation registration bringing bad luck, respect for the corpse and the need to keep it intact, and fear about the organ recovery procedure were mentioned.

(page 14, line 267-274)

Fear about the surgical recovery procedure may be rooted in misunderstandings about deceased organ donation [13–15, 17]. The belief that one’s body should remain intact after death is common in Asian cultures [22, 23]. Further, shielding one’s parents’ bodies from harm is considered an expression of filial piety [24, 25]. Donating vital organs is considered to result in one’s inability to “survive” in the world after death [13, 24]. Some Buddhists believe that the soul needs at least 24 hours to integrate and prepare to leave a deceased body, and that harm done to the body after physical death can still be “felt” in the soul; therefore, during this time, the body must not be tampered with [26, 27].

(page 14-15, line 275-276)

When a person is alive, it is taboo to talk about the possibility of their subsequently becoming a deceased organ donor as it makes people uncomfortable.

(page15-16, line 298-299)

Notably, higher deceased organ donation rates have been observed in countries that implement the opt-out rather than the opt-in principle [28]

---

## [Decision Letter · Decision Letter 2]

8 Feb 2021

Factors associated with registration for organ donation among clinical nurses

PONE-D-20-30483R2

Dear Dr. Weng,

We’re pleased to inform you that your manuscript has been judged scientifically suitable for publication and will be formally accepted for publication once it meets all outstanding technical requirements.

Kind regards,

Nicola Lacetera

Academic Editor

PLOS ONE

Additional Editor Comments (optional):

Reviewers' comments:

Reviewer's Responses to Questions

**Comments to the Author**

1. If the authors have adequately addressed your comments raised in a previous round of review and you feel that this manuscript is now acceptable for publication, you may indicate that here to bypass the “Comments to the Author” section, enter your conflict of interest statement in the “Confidential to Editor” section, and submit your "Accept" recommendation.

Reviewer #1: All comments have been addressed

Reviewer #2: All comments have been addressed

2. Is the manuscript technically sound, and do the data support the conclusions?

Reviewer #1: Yes

Reviewer #2: Yes

3. Has the statistical analysis been performed appropriately and rigorously? 

Reviewer #1: Yes

Reviewer #2: Yes

4. Have the authors made all data underlying the findings in their manuscript fully available?

Reviewer #1: Yes

Reviewer #2: Yes

5. Is the manuscript presented in an intelligible fashion and written in standard English?

Reviewer #1: Yes

Reviewer #2: Yes

6. Review Comments to the Author

Reviewer #1: Dear Respectable authors

I reviewed the responses. Thank you for your clarification. In my opinion, you paper acceptable for publication.

Cheers

Reviewer #2: (No Response)

7. PLOS authors have the option to publish the peer review history of their article (what does this mean?). If published, this will include your full peer review and any attached files.

Reviewer #1: **Yes: **Morteza Arab-Zozani

Reviewer #2: No

---

## [Editor Report · Acceptance letter]

10 Feb 2021

PONE-D-20-30483R2 

Factors associated with registration for organ donation among clinical nurses 

Dear Dr. Weng:

I'm pleased to inform you that your manuscript has been deemed suitable for publication in PLOS ONE. Congratulations! Your manuscript is now with our production department. 

Kind regards, 

on behalf of

Professor Nicola Lacetera 

Academic Editor

PLOS ONE